# Novel Insight of Histamine and Its Receptor Ligands in Glaucoma and Retina Neuroprotection

**DOI:** 10.3390/biom11081186

**Published:** 2021-08-11

**Authors:** Silvia Sgambellone, Laura Lucarini, Cecilia Lanzi, Emanuela Masini

**Affiliations:** 1Department of Neuroscience, Psychology, Drug Research and Child Health (NEUROFARBA), Pharmacology and Toxicology Section, University of Florence, 50139 Florence, Italy; silvia.sgambellone@unifi.it (S.S.); emanuela.masini@unifi.it (E.M.); 2Toxicology Unit, Emergency Department, Careggi University Hospital, 50139 Florence, Italy; lanzic@aou-careggi.toscana.it

**Keywords:** histamine, intraocular pressure (IOP), histamine H_3_R antagonists, baro-protection

## Abstract

Glaucoma is a multifactorial neuropathy characterized by increased intraocular pressure (IOP), and it is the second leading cause of blindness worldwide after cataracts. Glaucoma combines a group of optic neuropathies characterized by the progressive degeneration of retinal ganglionic cells (RGCs). Increased IOP and short-term IOP fluctuation are two of the most critical risk factors in glaucoma progression. Histamine is a well-characterized neuromodulator that follows a circadian rhythm, regulates IOP and modulates retinal circuits and vision. This review summarizes findings from animal models on the role of histamine and its receptors in the eye, focusing on the effects of histamine H_3_ receptor antagonists for the future treatment of glaucomatous patients.

## 1. Introduction

Glaucoma is a group of optic multifactorial neuropathies characterized by the progressive degeneration of retinal ganglionic cells (RGCs). These are central nervous system (CNS) neurons with cell bodies in the inner retina and axons in the optic nerve. Degeneration of the optic nerve results in progressive, permanent vision loss, starting with hidden blind spots at the edges of the visual field, progressing to tunnel vision, and finally to blindness [1]. The biological basis of glaucoma is poorly understood, and the factors contributing to its progression have not been fully characterized [2].

Glaucoma represents a significant public health problem. It is the second leading cause of blindness worldwide after cataracts, and the global prevalence of glaucoma in the population aged 40 to 80 years is 3.5% [3].

There are different types of glaucoma: hypertensive and normotensive glaucoma are the most common. In the first, elevated intraocular pressure (IOP) is caused by an imbalance between the production and outflow of aqueous humors (AH). In the second, the optic nerve head becomes damaged even though IOP is within the normal range. We assist with a reduction in ocular perfusion through less oxygen and nutrients supplied to the optic nerve. Physiologic IOP is generally within a range between 10 and 20 mmHg to ensure constant corneal curvature. This pressure is determined by the volume of AH within the eye exerting an outward pressure, scleral compliance and extraocular muscle tone exerting inward pressure [4].

The increased intraocular pressure with altered IOP circadian waves, if untreated, can reduce local perfusion, recurrent ischemic insults, and ultimately reduces the number of RGCs by autophagy and apoptosis [5]. When the number of RGCs is no longer suitable for neural transmission, the visual field becomes more and more narrow. Several mechanisms have been proposed to explain nerve head damage in glaucoma. Elevation of IOP and axial shallowing of the anterior chamber from posterior pressure characterize AH misdirection; however, the precise mechanism of increased resistance to AH outflow remains unclear, and is currently an active focus of research [6,7,8,9].

The histaminergic system in the eye may exert modulatory functions on visual activity and modulate IOP homeostasis and circadian fluctuations. Along with other factors, the dysregulation of histamine secretion may thus contribute to the pathophysiology of glaucoma.

## 2. Mechanism of Damage of Optic Nerve

Raised IOP is thought to damage the optic nerve head via induced mechanical changes at the lamina cribrosa or via vascular dysfunction and resultant ischemia. Several mechanisms are postulated to cause elevated IOP, the majority of which are related to reduced AH outflow. Structural changes include:Trabecular meshwork obstruction by foreign material (e.g., glycosaminoglycans, pigments, red blood cells);Trabecular cell loss of phagocytic activity and death;Loss of giant vacuoles from Schlemm’s canal endothelium and reduced pore size and density in the wall of this canal.

These changes could be brought on by altered endogenous corticosteroid metabolism with secondary trabecular meshwork changes and IOP rise [10].

Following isolation of the *Myocilin (MYOC)* gene, the clinical manifestation of glaucoma has been associated with a genetic predisposition. This gene, previously known as the *trabecular meshwork inducible glucocorticoid response* (*TIGR)* gene [11], was identified in pedigrees with juvenile open-angle glaucoma (OAG), a term used to refer to primary open-angle glaucoma (POAG) earlier age at onset, and an autosomal dominant mode of inheritance [12]. In recent years, the role of the *MYOC* gene has been extensively studied. The sequence variations in the gene account for approximately 2% to 4% of glaucoma cases. One particular *MYOC* mutation, Gln368Stop (dbSNP accession number: rs74315329), is the most common genetic mutation, causing glaucoma by increasing IOP [13].

It has recently been demonstrated that the integrity of Schlemm’s canal is maintained by coordinated functions of angiopoietin-Tie2 (Angpt-Tie2) signaling, and *TIE2* mutations have been identified in patients with POAG [14]. The baro-protection remains the most important therapeutic approach for preserving visual function in glaucoma patients; however, in recent years, a normal retinal and choroidal vascular perfusion and neurotrophic factor preservation have arisen as important [15].

Multiple treatments for reducing ocular hypertension and glaucoma exist and are chiefly separated into pharmacological, laser, and surgical therapies [10,16,17]. The pharmacological treatment of glaucoma includes drugs of various classes, generally administered as eye drops: prostaglandin analogues, β-blockers, carbonic anhydrase inhibitors, α-adrenergic agonists, miotics, hyperosmotic agents, and recently, Rho-kinase inhibitors [18]. These drugs can be administered singularly or in association, but a significant number of patients are “*non-responders*” to these treatments even when they are administered in combination. The therapeutic options available for lowering IOP, especially relating to the enhancement of AH outflow through the trabecular pathway, are limited, and in addressing this challenge, novel therapeutic approaches have been the focus. The current review presents an overview of the role of histamine in glaucoma and the future use of histaminergic drugs to treat glaucomatous patients.

## 3. The Histaminergic System

In contrast with other biogenic amines, histamine has garnered less attention because of its moderate action in the CNS. However, recent evidence suggests that histamine plays an essential role in multiple neuronal disorders, including insomnia, narcolepsy, Parkinson’s diseases, schizophrenia, Alzheimer’s disease, and cerebral ischemia [19,20]. In the CNS, histamine is produced within mast cells and neurons [21]. Histidine decarboxylase (HDC) is the rate-limiting enzyme in the formation of histamine, catalyzing the synthesis of histamine from the amino acid L-histidine. Within the CNS, the tuberomamillary nucleus of the hypothalamus is the leading site for the neuronal synthesis of histamine. Four histamine receptors (H_1_R–H_4_R) have been identified, and these receptors induce neuronal effects via G-protein coupled signaling mechanisms.

Histamine H_1_ receptor (H_1_R) is ubiquitously expressed, particularly in lungs, the CNS and blood vessels. It couples to Gαq/11 proteins, triggering phospholipase C (PLC) and protein kinase C (PKC) activation, together with inositol-1,4–5-trisphosphate (IP_3_) formation and intracellular Ca^2+^ release [22].

Histamine H_1_R activation can provoke a type I allergic reaction, whose typical signs are pruritus, increased vascular permeability and edema. Consequently, the administration of H_1_R antagonists (antihistamines) is the most important anti-allergic therapeutic intervention [23]. In the CNS, H_1_R is involved in cognitive functions, locomotor activity, emotions, arousal, circadian rhythm, sleep, or pain perception [24]. The most important side effect of brain-penetrating first-generation antihistamines is sedation, and it is caused by antagonism at H_1_R in the CNS [23,25].

Histamine H_2_R is ubiquitously expressed, particularly in the stomach, heart, and CNS [22,25]. In 1977, Sir James Black’s research group released the H_2_R antagonist cimetidine as the first efficient pharmacological treatment for gastroesophageal reflux disease, dyspepsia, gastric or duodenal ulcers. Agonist binding to H_2_R results in the activation of Gαs-proteins that stimulate the adenylyl-cyclase-mediated production of the second messenger cAMP. In the brain, functions of the H_2_R include the modulation of cognitive processes, circadian rhythm, food intake and glucose metabolism [24]. Experiments with *knockout* mice have shown that the histamine H_2_R is implicated in modifying the immune responses, specifically in the modulation of Th1- or Th2-cell polarization [26].

Histamine H_3_R is primarily expressed in neurons and acts as a presynaptic auto- and hetero-receptor. It inhibits the release of histamine [27,28] and neurotransmitters, such as acetylcholine, dopamine, noradrenaline or glutamate [29]. Histamine H_3_R is considered a potential target for treating cerebral disorders [30]; furthermore, it is expressed post-synaptically in the basal ganglia and within the dorsal and ventral striatum [31]. Histamine H_3_R is coupled with G_i/o_ protein. It inhibits adenylyl cyclase and the high-voltage-activated Ca^2+^-channels, which regulate histamine synthesis and neurotransmitter release. The inverse H_3_R agonist pitolisant is currently used to treat narcoleptic patients [32].

Histamine H_4_R, discovered in 2000, is the most recently identified histamine receptor. This discovery culminated with six independent groups reporting the cloning of the histamine H_4_R [22]. This receptor modulates the migration and activation of a wide spectrum of immune cells (basophils, mast cells, eosinophils, dendritic cells, monocytes, NK, iNK T and γδ cells, CD8^+^ T cells, Treg, and Th2 cells), and it is thus involved in allergic and immune-mediated disorders.

Histamine H_4_R antagonists prevent mast cell chemotaxis and submucosal accumulation in the trachea of mice after histamine inhalation [33], lung inflammation and fibrosis induced by bleomycin [34,35]. This receptor is expressed in the hippocampus, granular layer of the cerebellar cortex, thalamus, and spinal cord [36]. Recently, observations conducted on histamine H_4_R *knockout* mice [37] have shown that H_4_R modulates locomotor activity, anxiety, nociception and feeding behavior, confirming the role of the histaminergic regulation of neuronal functions and in neuropathic pain [38]. Dysregulation of brain histamine is a potential contributor to neuropsychiatric [29] and Parkinson’s [19] diseases, as well as in Tourette syndrome, a rare genetic disease [39].

## 4. The Histaminergic System at Ocular Level

Retina, optic nerve and various brain structures of albino and pigmented rabbits contain histamine in the range of 40–400 ng/g of tissue; choroid tissue of both animal strains is characterized by amine contents several times higher. In the retina, no histamine-forming cells have been identified to date; however, retinopetal axons arising from the tuberomamillary nucleus extend across the inner plexiform layer, eliciting responses in a range of inner retinal neurons [40]. The synthesis and release of histamine are controlled by presynaptic histamine H_3_ auto-receptors located in the CNS [27,28]. Histamine H_1_, H_2_ and H_3_R have been localized in the inner layer of ganglion cells in rodents and primate retinae [40,41]; in particular, circuits involved in scotopic vision may be altered by histamine release. In the macaque retina, the stimulation of histamine H_3_R increases the delayed rectifier component of the voltage-dependent potassium conductance in ON bipolar cells [42], and in dark-adapted baboon retinas, histamine decreases the rate of maintained firing and the amplitude of the light responses of ON ganglion cells [43]. The primate retina receives input from histaminergic neurons that are active during the day in the posterior hypothalamus, and they receive input from the brain via axons emerging from the optic nerve. One set of retinopetal axons arises from perikarya in the posterior hypothalamus and uses histamine, whereas in the dorsal raphe serotonin is utilized. These histaminergic and serotonergic neurons are not specialized to supply the retina; they are neurons’ subsets that project via collaterals to many other targets in the CNS. They are components of the ascending arousal system, active when the animal is awake. Many of the effects of histamine and serotonin on light responses suggest that retinopetal axons optimize retinal function at an ambient light intensity during the waking period.

Histamine activates chloride channels, increasing chloride conductance, in the monopolar optic cells in insects, suggesting a role of histamine in photoreceptor response to light in flies [44]; moreover, severe mutation in the gene encoding for histamine receptors causes a defect in the transients of the electroretinogram derived from large monopolar cells of the first optic lamina [45]. Histamine reduces the amplitude of light responses in monkey RGCs, a finding consistent with a role for retinopetal axons in light adaptation in these diurnal animals [46]. These findings suggest that histamine acts primarily via volume transmission in the primate retina, increasing the operating range of cones and conserving ATP in bright, ambient light. The histaminergic system is deeply implicated in circadian rhythm and fulfils a significant role in maintaining waking [47]. During the day, the histaminergic tone could play a role in maintaining the IOP balance. Furthermore, histamine is responsible for ciliary muscle contraction in human eyes and, therefore, in IOP reduction [48].

Ocular administration of histamine triggers local inflammation in a non-specific manner; the severity of conjunctivitis is dose-dependent; however, histamine is well tolerated, although transient blepharitis, aqueous flare, and ocular hypertension occur in some experimental situations [49]. Total protein content and serum albumin levels increase after histamine administration, as are lacrimal albumin levels during naturally acquired conjunctivitis; lacrimal albumin concentration decreases in parallel with the reduction in the conjunctivitis score [49]. In the conjunctival immediate hypersensitivity reaction (type I allergy), histamine is released from degranulated mast cells in the early and late phase [50]; histamine levels in the tears of kerato-conjunctivitis (KCV) patients are higher than in control subjects, and histamine H_1_ receptors are over-expressed in the active phase of the disease [51].

Histamine H_1_Rs are localized on horizontal cells and in a small number of amacrine cells, whereas histamine H_2_Rs appear closely associated with synaptic ribbons inside cone pedicles [41]. Histaminergic H_1_ and H_2_ receptors in the iris arterioles and H_2_ receptors in the iridal venules modulate vascular tone in rats [52]. Moreover, several ocular hypertensive effects have been reported in chronic glaucoma patients following the use of cimetidine and ranitidine, two H_2_R antagonists, for peptic ulcer treatment [53]; on the contrary, recent studies have failed to demonstrate the significant action of topical administered H_2_ blockers on IOP in humans [54]. Histamine H_1_ and H_2_R antagonists possess anticholinergic activity that may induce glaucoma. Promethazine, an antipsychotic with antihistamine activity, has been shown to produce an idiopathic swelling of the lens that could increase the risk of angle-closure glaucoma. Topical administration of ranitidine produces vasoconstriction in both the arterioles and the venules of the iris, suggesting a predominant role of histamine H_2_R in the vasculature of the iris [52].

Histamine H_4_ receptors are expressed by various inflammatory cells, including eosinophils and Th2 cells, in allergic disorders [55], and infiltrating inflammatory cells in subconjunctival tissues of KCV patients strongly express H_4_R [56]. H_4_Rs are mainly expressed in immune cells; thus, their down-regulation may lead to a decreased eosinophil infiltration into the conjunctival tissue. Therefore, the expression level of H_4_R on the ocular surface may be a useful biomarker for atopic KCV in clinical examinations [51].

Histamine tone is decreased at night, and nocturnal IOP is higher than diurnal pressure. IOP results from a balance between secretion and outflow of AH [57]. This balance is entirely controlled in healthy subjects, with three crucial mechanisms: the rate of AH formation, the resistance to the outflow, and episcleral venous pressure. These factors change during the night, the AH production decreases significantly in diurnal mammalians, as does the drainage; thus, the IOP increases [58]. Histamine regulates ciliary muscle contraction in human eyes and, therefore, it controls IOP reduction [48]. Moreover, AH production or outflow rate can be influenced by histamine, which is the neurotransmitter of the ascending arousal system, and the contributions of retinopetal axons to visions can be predicted from the known effects of histamine on retinal neurons.

## 5. The Role of Histamine H_3_ Receptors in the Control of Intraocular Pressure

The major drainage pathway of AH consists of structures located in the angular region of the anterior chamber of the eye, such as the trabecular meshwork and the Schlemm’s canal system, which are responsible for about 90% of outflow from the eye. Some accessory pathways include the uveoscleral system, with a small contribution of trans-corneal and vitreal flux [59]. Obstruction in the circulatory pathway of AH causes IOP elevation, which is a significant risk factor for glaucoma. AH production and outflow rate can be influenced by histamine. Moreover, histamine has effects on retinal neurons [46]. Previous work in our laboratory [60] reported the expression pattern of histaminergic receptors in rabbit eyes. In this work, histamine H_1_R and H_4_R expression were found in the retina and optic nerve at a higher concentration than that revealed in the trabecular meshwork and stomach, used as a positive control (Figure 1A,C). High histamine H_3_R protein expression levels were found in the retina, optic nerve and ciliary body (Figure 1B), whereas histamine H_2_R was found only in the stomach, which resulted in undetectable protein in the ocular tissues.

Several histamine H_3_R antagonists have been tested in animal models of glaucoma in rabbits. Both imidazole (ciproxifan) and non-imidazole compounds, such as DL-76 (1-[3-(4-tert-butylphenoxy)propyl]piperidine hydrogen oxalate, [60]) and GSK189254, at an equimolar concentration (1%) reduced IOP following a single acute challenge or after repeated doses in transient or stable IOP raise models in rabbits [60]. After 50 µL of 5% hypertonic saline injection in the eye’s anterior chamber, IOP increased from 16.8 ± 5.6 mmHg to 39.63 ± 4.85 mmHg. This value remained stable until 120 min, and decaying after that to reach baseline values at 240 min. All the compounds reduced IOP in a statistically significant manner with a different profile; ciproxifan and DL-76 were more effective than GSK189254 (Table 1A). As observed in the transient ocular hypertensive model, all the compounds, at a 1% dose, caused a significant reduction in IOP in the carbomer-induced chronic model in male New Zealand White (NZW) rabbits after seven days of treatment. Ciproxifan and DL-76 were the most effective compounds. The effect of timolol at a 1% dose, the gold standard treatment, is also reported (Table 1B).

Interestingly, the IOP-lowering activity of ciproxifan and DL-76 were suppressed by the pre-treatment with 1% imetit, a histamine H_3_R agonist, providing the specificity of histamine H_3_R antagonism action (Figure 2).

The effect of histamine H_3_R antagonists on the vascular performance at the posterior pole of the eye was evaluated through Doppler ultrasound studies of the ophthalmic artery. In elevated IOP eyes, the Pourcelot Resistivity Index (RI) was significantly higher compared to physiological IOP eyes [61]; the treatment with histamine H_3_R antagonists significantly reduced Pourcelot RI, indicating a role of histamine in controlling the vascular tone at the ocular level. It is well known that vascular impairment in the eye is important in the pathogenesis of normotensive glaucoma. Hence, chronic treatments with H_3_R antagonists could have a significant role in improving this kind of glaucoma [60]. The IOP control with a chronic treatment with H_3_R antagonists significantly prevented the cell death of neurons in the RGC layer of hypertensive rabbit eye (Figure 3). Moreover, RGC death is the leading cause of visual impairment in glaucomatous patients.

In conclusion, ocular hypertension and IOP fluctuation are the most critical risk factors in glaucoma progression. Histamine receptors are expressed in a neuronal and non-neuronal compartment in the eye, as well as in trabecular-meshwork-derived cells, and histamine H_3_R antagonists are effective in reducing IOP, controlling ocular vascular performance and preventing RGC death.

Recent studies have identified the importance of endothelial signaling molecules in maintaining the AH conventional outflow pathway and roles for Tie2, a tyrosine kinase receptor located on vascular endothelial cells and its ligands angiopoietin-1 and-2 (Angpt-1 and Angpt-2), have been described. In patients with primary congenital glaucoma, a severe form of the disease characterized by an early onset and a severe neuropathy, a mutation of the gene encoding Angpt-1 has been shown [14]. The deletion of *ANGPT-1* and *-2* or *TIE2* genes in adult mice severely impaired the integrity of Schlemm’s canal, leading to IOP elevation, retinal neuron damage and RGC death, all features of glaucoma in humans [62]. The actions of histamine and H_3_R antagonists are present at the vascular level; therefore, an interaction between histamine and the Tie2 receptor could be postulated for the control of IOP and glaucoma progression.

In conclusion, the histaminergic system may represent a new site for multiple therapeutic interventions that could hamper the progression of glaucoma damage and visual field loss in patients and histamine H_3_R antagonists, a new class of IOP lowering agents with an innovative mechanism of action.

## Figures and Tables

**Figure 1 biomolecules-11-01186-f001:**
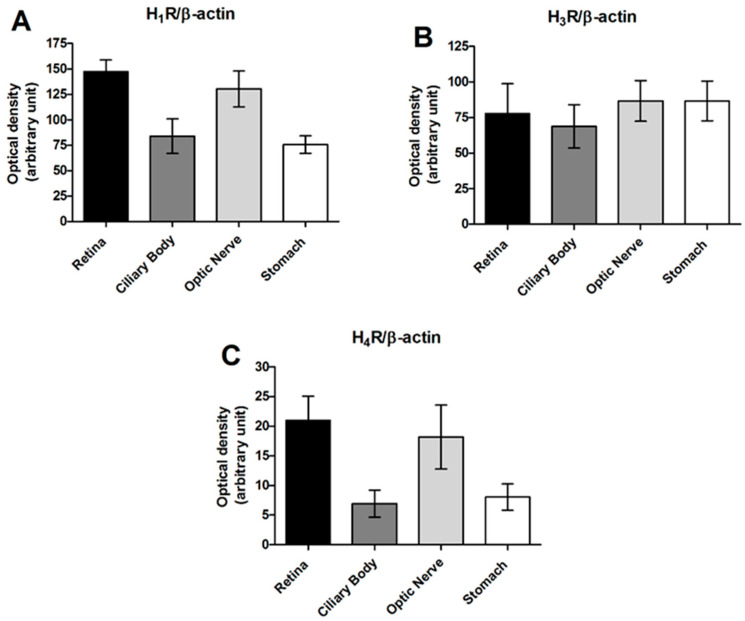
Western blot analysis of histamine H_1_R (**A**), H_3_R (**B**) and H_4_R (**C**) subtypes in the retina, ciliary body, optic nerve, and stomach samples of New Zealand White (NZW) rabbits. Densitometric data of 6 determinations are reported as relative optical density, corrected for the corresponding β-actin content [60] modified.

**Figure 2 biomolecules-11-01186-f002:**
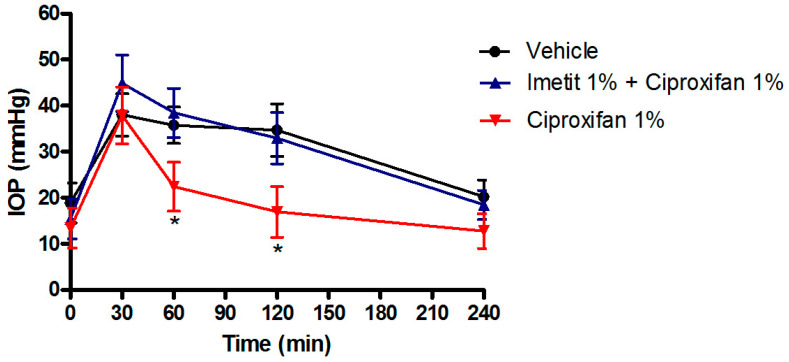
IOP lowering effect of ciproxifan in the transient ocular hypertensive model in NZW rabbits. The effect of ciproxifan is suppressed by pre-treatment with 1% imetit. * *p* < 0.05 ciproxifan 1% at 60′ and 120′ vs. vehicle and imetit 1% + ciproxifan 1%. All the results are expressed as mean ± SEM (*n* = 6). Two-way ANOVA followed by Bonferroni post hoc test.

**Figure 3 biomolecules-11-01186-f003:**
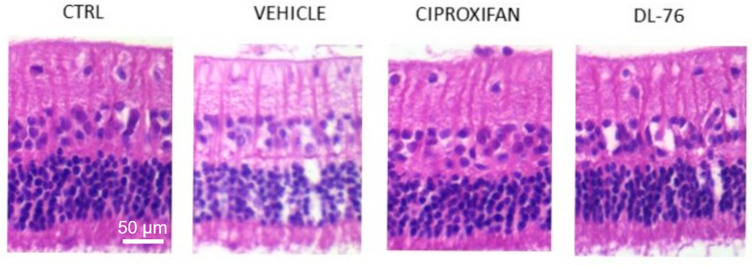
Representative images of hematoxylin/eosin-stained histological sections of retinae from different treated groups. RGCs are visible in the upper layer. The histological sections of Vehicle, Ciproxifan and DL-76 panels were prepared from carbomer-induced glaucoma models in male NZW rabbits with stable elevated IOP [60] modified.

**Table 1 biomolecules-11-01186-t001:** (**A**) Transient ocular hypertensive model and (**B**) carbomer-induced glaucoma model in male NZW rabbits.

**A**		**IOP-Lowering Effect**				
**Compound**	**Basal**	**After** **Saline**	**Post Treatment (60 min)**	**ΔΔIOP** **(60 min)**	**Post Treatment (120 min)**	**ΔΔIOP** **(120 min)**
	IOP, mmHg	IOP, mmHg	IOP, mmHg	mmHg	IOP, mmHg	mmHg
Vehicle	15 ± 0.3	36 ± 7.5	38 ± 3.9	0 ± 3.9	28 ± 5.7	0 ± 5.7
Ciproxifan	13 ± 4.3	38 ± 6.2	22 ± 5.3	−18.9 ± 5.3 **	17 ± 5.5	−16.4 ± 5.5 **
DL-76	15 ± 4.3	34 ± 6.2	23 ± 7.4	−15.4 ± 7.5 **	20 ± 3.0	−16.1 ± 3.1 **
GSK189254	14 ± 5.6	37 ± 5.5	32 ± 4.3	−8.5 ± 4.3 *	25 ± 6.7	−9.9 ± 6.7 *
Timolol	14 ± 2.5	38 ± 5.7	21 ± 3.8	−16.5 ± 3.8 **	19 ± 4.6	−14.8 ± 4.7 **
**B**		**IOP-Lowering Effect**				
**Compound**	**Basal**	**After** **Carbomer**		**Post-Treatment** **(7 days)**	**ΔΔIOP** **(7 days)**	
	IOP, mmHg	IOP, mmHg		IOP, mmHg	mmHg	
Vehicle	15 ± 0.3	38 ± 2.8		41 ± 7.3	0 ± 2.12	
Ciproxifan	13 ± 5.1	36 ± 4.1		20 ± 2.9	−19 ± 2.9 **	
DL-76	12 ± 3.5	34 ± 2.8		23 ± 3.8	−15.7 ± 2.7 *	
GSK189254	14 ± 0.0	40 ± 1.4		26 ± 2.8	−14.5 ± 3.8 *	
Timolol	15 ± 0.7	41 ± 5.6		27 ± 2.1	−13.5 ± 2.1 *	

Ocular hypotensive efficacy is expressed in mmHg. The average difference in IOP between drug-treated eyes or vehicle-treated eyes and their respective pre-treatment value are shown in the following formula: Efficacy = (IOP_drug_ − IOP_predose drug_) − (IOP_veh_ − IOP _predose veh_). (**A**) ** *p* < 0.01 ciproxifan, DL-76 1% and Timolol 1% at 60′ and 120′; * *p* < 0.05 GSK189254 1% at 60′ and 120′ versus vehicle; (**B**) ** *p* < 0.01 ciproxifan; * *p* < 0.05 DL-76, Timolol, GSK189254 at day 7 versus vehicle. All the results are expressed as mean ± SEM (*n* = 6). The significance of differences was assessed by two-way ANOVA for multiple comparisons followed by the Bonferroni post hoc test [60] modified.

## Data Availability

Data are available on request.

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
