# Peer review of "Novel Insight of Histamine and Its Receptor Ligands in Glaucoma and Retina Neuroprotection"

_biomolecules, 2021, doi:10.3390/biom11081186_

Round 1
Reviewer 1 Report
Abstract
Line 4 “and it is the second leading…..after cataract” should be deleted. Because this sentence is repeated.
It is better to show IOP time course in Table 1B.
Line 231 “Figure 1, A and B” should is changed “Figure 1, A and C”.
Line 233 “Figure 1, C” should is changed “Figure 1, B”.
Scale bar should be added in Figure 3.
If the authors evaluate the number of RGCs, flat retinal mountain must be shown. It is not enough to evaluate RGCs in the retinal section.
Reviewer 2 Report
This review describes the evidences mainly from animal experiments that indicate H3 receptor as a novel therapeutic target of glaucoma. It covers the roles of HA system in retinal cells and optic fibers as well as IOP regulation, which is a crucial element for glaucoma pathophysiology and treatment. This review offers new concept of glaucoma treatment, however before publication, I would like to encourage the authors to revise some minor parts.
Remarks:
- Among glaucoma, normal tension glaucoma (NTG) accounts for about 50% of glaucoma in Asia (Am J Ophthalmol. 2019 Mar;199:101-110. doi: 10.1016/j.ajo.2018.10.017.) and population-based studies have found that from 10% to 48% of all open angle glaucoma patients in the United States, Europe and Scandinavia have NTG. There is a hypothetic pathophysiology of NTG that suggest perfusion deficit and vascular dysregulation in the retina are involved in retinal ganglion cell loss (Eye (Lond). 2018 May;32(5):924-930. doi: 10.1038/s41433-018-0042-2.). Could the authors discuss the therapeutic potentials of H3 antagonists and other HA related drugs in the NTG?
- Brief description of physiological IOP regulation would be needed to understand the mechanism of increased IOP and optic nerve damage.
- Is there any correlation between IOP and the central HA circadian changes? Does the fluctuation of plasma HA concentration affect IOP circadian rhythm?
- The authors well described the role of HA system in retinal cells. Is there any direct contribution of HA system to improve damaged retinal functions in glaucoma?
- The transition of H3 antagonists into the CNS should be discussed to evaluate the adverse effects of eyedrop administration.
- line 93: tuberomammillary or tuberomamillary, which is correct? (Nat Rev Neurosci 4, 121–130 (2003). https://doi.org/10.1038/nrn1034)
Minor comments
- Please distinguish between gene and protein precisely. Human gene should be described Italic Capital.
- Statistical data should be informed in Table 1. Please offer reference of Table 1 too.
- lines 143-227: It would be appreciated if you could prepare the table that show eye tissue (retina and aqueous humor system), expressed histaminergic molecules (HA and receptors) and the spices (e.g. Mus musculus) , which are already described in the text.
